# Some Examples of Bacterial Toxins as Tools

**DOI:** 10.3390/toxins16050202

**Published:** 2024-04-23

**Authors:** Gudula Schmidt

**Affiliations:** Institute of Experimental and Clinical Pharmacology and Toxicology, University of Freiburg, Albertstr. 25, 79104 Freiburg, Germany; gudula.schmidt@pharmakol.uni-freiburg.de

**Keywords:** bacterial toxin, tool, protein delivery

## Abstract

Pathogenic bacteria produce diverse protein toxins to disturb the host’s defenses. This includes the opening of epithelial barriers to establish bacterial growth in deeper tissues of the host and to modulate immune cell functions. To achieve this, many toxins share the ability to enter mammalian cells, where they catalyze the modification of cellular proteins. The enzymatic activity is diverse and ranges from ribosyl- or glycosyl-transferase activity, the deamidation of proteins, and adenylate-cyclase activity to proteolytic cleavage. Protein toxins are highly active enzymes often with tight specificity for an intracellular protein or a protein family coupled with the intrinsic capability of entering mammalian cells. A broad understanding of their molecular mechanisms established bacterial toxins as powerful tools for cell biology. Both the enzymatic part and the pore-forming/protein transport capacity are currently used as tools engineered to study signaling pathways or to transport cargo like labeled compounds, nucleic acids, peptides, or proteins directly into the cytosol. Using several representative examples, this review is intended to provide a short overview of the state of the art in the use of bacterial toxins or parts thereof as tools.

## 1. Pore Forming Bacterial Toxins

A large number of bacterial toxins do not have catalytic activity but instead form pores in the plasma membrane by oligomerization [1]. Water-soluble monomers are released by pathogenic bacteria, bind to cellular receptors (lipids, glycolipids, glycoproteins, or proteins), and are incorporated into the membrane for oligomerization. This is accompanied by a remarkable conformational change, leading to the formation of water-filled pores, changes in the membrane potential, and eventually cell death [2]. Pore-forming toxins are grouped into two structurally different classes. Prominent examples of alpha pore-forming toxins (the membrane channel is created by membrane-spanning alpha helices) are the family of Cytolysins (Cly A), which target mammalian cells, and the Colicins, which act on other bacteria for competition with food sources (Table 1) [3]. Together with aerolysins and hemolysins, the cholesterol-dependent cytolysins (CDC) belong to the family of β-pore forming toxins. They consist of amphipathic β-strands [4,5] and form pores between 1 nm and 100 nm in diameter [6]. Pore formation itself and changes in the cytosolic ion composition induce survival mechanisms of the targeted cells. For example, the shedding of pore-containing membrane vesicles is induced by the influx of calcium ions. Additionally, endocytosis and lysosomal degradation of the protein toxins as well as activation of autophagy have been described [7].

Stable pores also form in model bilayer membranes called black lipid membranes. This allows the determination of the biophysical properties including ion selectivity and the size of the channels formed [8,9]. Particularly, because of the non-permeable nature of the plasma membrane, toxins that form big pores, like streptolysin O, can be used to load small molecules and even peptides or drugs into the cytoplasm of mammalian cells.

Although streptolysin O treatment is temporally harmful to the target cell, channel formation seems to be reversible. Indeed, it has been shown that several cell lines survive toxin treatment in cell culture experiments [10]. However, the transport of ions or soluble small molecules through water-filled pores is exclusively driven by diffusion (Figure 1). This is different when toxins are used, which form pores aiming at the selective transport of an enzyme into the cytosol of target cells (see Section 2). Besides pore formation, the property of specific binding to cellular receptors has been utilized. For example, *Clostridium perfringens* Perfringolysin O (PFO) binds to cholesterol. Domain 4 of PFO, which is sufficient for this interaction, has been used as a biosensor for cholesterol nanodomains in cell biology [4,11]. Similarly, non-pore-forming toxins have been used for specific labeling of cellular structures. *Vibrio cholerae* cholera toxin binds to the ganglioside GM1 and *Shigella dysenteriae* Shiga toxin binds to the glycolipid Gb3. Both toxins have been utilized to monitor lipid trafficking within cells [12,13].

## 2. Toxins Forming Pores for Delivery of an Enzyme

Some toxins form pores with the objective of transporting toxic enzymes into mammalian cells. For this purpose, the enzymatically active part (A) binds to the pore formed by a second toxin component (B). The most prominent example of such a two-component AB-type toxin is anthrax toxin [14]. The toxin produced by *Bacillus anthracis* is composed of a protective antigen (PA), which binds to the surface of mammalian cells, and of two separate enzymes: lethal factor (LF) and edema factor (EF). PA specifically interacts with tumor endothelial marker 8 (TM8, anthrax toxin receptor 1) or with capillary morphogenesis gene 2 (CMG2, anthrax toxin receptor 2) [15,16]. Oligomerization of PA leads to the formation of a heptameric prepore [17]. This process requires cleavage of each PA monomer by the abundant serine protease furin or furin-like proteases on the cell surface [18]. This property has been utilized for toxin engineering. The alteration of the cleavage site in PA toward target sequences for other proteases has been shown to modulate the specificity of the mutated toxin. This allowed turning the toxin’s activity toward cancer cells on which some proteases are highly expressed [19]. Moreover, a mutant of PA that lost its affinity to the natural receptors was re-targeted to other membrane proteins. For example, fusion of EGF or an Her2 affibody to the mutated PA enhanced the specificity of the engineered anthrax toxin pore for EGFR/Her2 expressing tumor cells by the factor of 100 [20,21,22].

Cleavage and subsequent prepore formation allow the binding of LF and/or EF. The protein complex is then taken up by clathrin-dependent endocytosis [23]. Upon acidification of the maturing endosome, the anthrax toxin pore inserts into the endosomal membrane and thereby facilitates the release of LF/EF into the cytosol. Cellular chaperones enable the folding of the proteins to their catalytically active conformation. Lethal factor is a metalloprotease that catalyzes the cleavage of the mitogen-activated protein kinase kinase (MAPKK) and Nlrp1, thereby influencing inflammasome formation. Edema factor enhances the amount of intracellular cAMP by acting as adenylyl cyclase (for review see [24]). Biochemical and structural analysis of anthrax toxin led to the identification of the domains of EF and LF that are essential for PA binding. The fusion of these domains with several protein cargos of choice allowed the transport of foreign proteins through the membrane channel formed by anthrax protective antigen. Besides the endogenous adapter domain of LF/EF, an artificial tag composed of positively charged amino acids was shown to be sufficient for binding to the PA oligomer and for mediating cargo delivery [25]. The combination of all these modifications lastly enabled the use of anthrax toxin for tumor targeting and drug transport into the cytosol. Anthrax toxin-based therapeutic applications have already been established in animal models [26].

Similar to anthrax toxin, *Clostridium botulinum* C2 toxin is a binary AB-type protein toxin consisting of a pore-forming unit (C2-II) and a second protein with enzymatic activity (C2-I), which is transported through the pentameric channel. Like PA, C2-II has to be cleaved to form the heptameric prepore and to bind C2-I. The protein complex is taken up by receptor-mediated endocytosis. The C2 receptor seems to be a common protein-linked carbohydrate, which complicates re-targeting the toxin to specific cells [27]. Acidification induces channel insertion into the endosomal membrane, allowing the release of C2-I into the cytosol [28]. C2-I is an ADP ribosyltransferase that modifies actin leading to the breakdown of the actin cytoskeleton and cell rounding. Comparable to anthrax toxin, the N-terminal part of C2-I or a polycationic tag sufficiently mediates the binding and transport of cargo proteins through the C2-II pore [29]. Using this method, the C2 toxin also has been engineered to deliver proteins into mammalian cells [30].

It must not be omitted that such pore formation coupled with protein transport across biological membranes is not exclusive for bacterial toxins but is also implemented by mammalian cells. Cytotoxic T lymphocytes release perforins together with granzymes to destroy virus-infected cells and thereby limit reproduction and spreading of the virus [31]. Perforins form pores in the cell membrane through which the apoptosis-inducing protease granzyme can be delivered.

## 3. The Benefit of Understanding the Enzymatic Activity of Bacterial Toxins

Besides the binary AB-type protein toxins, other toxins with enzymatic activity are composed of at least three independently acting domains located within one single protein: the toxic enzyme, a cell binding domain that also mediates endocytosis, and a third domain allowing delivery of the catalytic part from the endosome into the cytosol. In contrast to the toxins described above, no stable pore is formed in artificial membranes and the question of how the enzymatic parts of those proteins escape the endosome is not yet answered. However, the fusion of cargo proteins to the N-terminal binding/translocation domain of Diphtheria toxin also facilitates the delivery of fluorescent proteins and enzymes [32,33].

The domain composition of single-chained toxins allowed the use of isolated domains for cell biological or even pharmacological purposes. The enzyme domain is usually very efficient and modifies specific proteins or protein families within mammalian cells. With the knowledge about the exact modification sites and the fate of the modified substrates, bacterial toxins have been utilized to answer cell biology questions. The cellular function of the respective target proteins and connected signaling pathways have been studied extensively. Several bacterial toxins with different enzymatic activities are well characterized and have been utilized to study mammalian signaling cascades.

### 3.1. ADP Ribosyltransferases

Prominent examples of ADP-ribosyl-transferases are *Vibrio cholerae* cholera toxin and *Bordetella pertussis* pertussis toxin. Both toxins are composed of a cell-binding pentamer and the catalytic subunit. They are transported in endosomes via the Golgi apparatus to the endoplasmic reticulum (ER). There, the catalytic domains are separated and lose their structure. Export to the cytosol is mediated by the ER-associated degradation (ERAD) pathway and the translocated enzyme is refolded in the cytosol. This uptake mechanism is called “long-trip” uptake to distinguish it from the “short-trip” translocation from acidified endosomes [34,35]. The toxins specifically modify alpha subunits of heterotrimeric G proteins to permanently activate (cholera toxin, Gα_s_) or inactivate (pertussis toxin, Gα_i_) them [36]. Accordingly, the toxins have been proven to be invaluable tools to study the function of these important signaling molecules.

A further example is *Clostridium botulinum* C3 toxin, which selectively modifies and inactivates the small GTPase RhoA, a main regulator of the actin cytoskeleton. Its inactivation by C3 leads to the breakdown of actin filaments and rounding of cells [37]. In cells of neuronal origin, Rho inactivation and Rho-independent effects by the toxin induce the outgrowth of neurites [38,39]; from this emerged the idea of utilizing C3 for the regeneration of neuronal connections after severing neurites due to injury [40]. C3 lacks a binding and translocation domain and therefore requires fusion to other membrane-crossing domains, such as cell-penetrating peptides, to enter most mammalian cells. Alternatively, the toxin can be injected manually. However, binding to the type III intermediate filament protein vimentin has been suggested to be sufficient for cell entry into damaged neurons [41].

*Corynebacterium diphtheriae* diphtheria toxin (DT) modifies elongation factor 2 (EF-2) by ADP-ribosylation. Modification of EF-2 blocks translation and protein synthesis in target cells, eventually leading to cell death [42]. The catalytic domain of DT has been fused to several other (not lethal) bacterial toxins as a powerful readout system during CRISPR/Cas9-based screens for the identification of cellular receptors [43,44,45]. In addition, DT has also been studied for pharmacological applications for over 50 years and several fusion proteins have been tested for cell specificity and activity. Indeed, two drugs based on DT have been approved: Ontak and Tagraxofusp. Ontak (denileukin diftitox) is an IL-2 peptide fused to a DT catalytic domain deployed to destroy T-cells, whereas Tagraxofusp is DT fused to an IL-3 peptide for selective removal of IL3 receptor-expressing cells [46]. The recombinant protein is established as a first-line therapy against the Blastic Plasmacytoid Dendritic Cell Neoplasm (BPDCN) [47,48].

An additional immunotoxin assembled from DT and single-chain fragments derived from an anti-PD1 antibody was recently developed. It targets programmed death 1 (PD1), a checkpoint receptor effectively depleting PD1-expressing cells in vitro and in mice [49]. An immunotoxin with similar activity but a lower yield of the recombinant protein was produced before anti-PD1-ABD-PE [50]. This fusion protein is based on a different toxin: the ribosyl-transferase *Pseudomonas aeruginosa* exotoxin A (ExoA). Like diphtheria toxin, ExoA targets an elongation factor and blocks protein synthesis [51]. Overall, one can state that engineered immunotoxins are on the way to becoming effective therapeutic agents.

### 3.2. The Deamidating Toxins CNF, PMT, and DNT

The AB toxins *Bordetella bronchiseptica* dermonecrotic toxin (DNT), *Pasteurella multocida* toxin (PMT), and the cytotoxic necrotizing factors (CNF1-3, CNFY), which are produced by pathogenic *Escherichia coli* strains and *Yersinia pseudotuberculosis*, encompass a family of deamidating AB toxins [52]. They catalyze the modification of Rho GTPases (DNT and CNFs) or alpha subunits of heterotrimeric G proteins (PMT). In each case, a specific glutamine of the target is deamidated to glutamic acid. This leads to persistent activation of the G proteins because the inactivation mechanism, the hydrolysis of the bound GTP, is blocked [53,54]. All toxins activate the complete pool of target proteins within mammalian cells and are frequently used to study Rho/Gα-dependent signaling pathways [55,56]. The toxins are taken up by receptor-mediated endocytosis. The T-type calcium channel Cav3.1 has been identified as a cellular receptor for DNT, [43]. PMT has been shown to interact with lipid components of the membrane [57,58] and to bind to the LDL-receptor-related protein 1 (LRP1) [59]. The receptors for CNFs have not been identified so far. All of these toxins are proteolytically cleaved within the endosome for release of the catalytic part into the cytosol. Blocking acidification of the endosomes by Bafilomycin A showed that the pH change is required for the transport of the catalytic domains into the cytosol, a fact known for many AB toxins [60,61,62].

*Pasteurella multocida* strains that produce PMT cause atrophic rhinitis in affected pigs. The toxin induces bone loss by inhibiting bone-building osteoblasts with simultaneous activation of bone-resorbing osteoclasts [63]. Moreover, PMT modulates the function of several immune cells [64]. In cultivated cells, PMT induces cytoskeletal rearrangements and acts as a strong mitogen. Besides unraveling signaling pathways downstream of G-protein coupled receptors, PMT has been studied as a possible pharmacological agent for treating fibrodysplasia ossificans, a rare human bone disease associated with massive pathological bone formation [65,66].

Activation of Rho proteins by *Escherichia coli* CNF1 leads to rearrangement of the actin cytoskeleton [67]. In contrast to the inactivating modifications described above, activation by CNF1 seems to be long-lasting. This is probably because there is no need for the cell to destroy an active protein. Although CNF1 is a toxin and induces necrotizing lesions when applied to the skin, it induces several beneficial changes in physiological responses when injected into tissues. For example, local administration of CNF1 into the footpads of mice enhanced analgesia [68]. Also, systemic application reduced pain by up-regulating µ-opioid receptors [69]. The toxin also promises beneficial properties against brain tumors. It promoted senescence and/or death of mouse and human glioblastoma cells [70]. Moreover, CNF1 induced the outgrowth of neurites and stimulated the morphogenesis of dendritic spines to modulate plasticity learning in a mouse model [71,72,73]. Both facts led to the attempt to inject the Rho activator directly into the liquor of glioma-bearing mice since gliomas and especially glioblastomas are extremely aggressive brain tumors. Indeed, the toxin enhanced neuronal function, decreased the tumor volume, preserved the healthy tissue, and increased the survival time of treated mice [74]. Recently, a fusion toxin was constructed encompassing chlorotoxin, a peptide from the venom of the scorpion *Leiurus quinquestriatus*, and CNF1. Adding the peptide enabled the protein to efficiently cross the blood–brain barrier. It allowed systemic application of the large protein while still showing anti-neoplastic activity in a glioblastoma mouse model [68,70]. Recently, CNF1 was also used to rescue cognitive dysfunction in a mouse model of Rett Syndrome [75]. Proceeding with these promising studies about the usefulness of CNF1 and other bacterial deamidases may pave the way for new and unexpected clinical applications.

### 3.3. Toxins with Other Enzymatic Activities

The most impressive existing application of a bacterial toxin is the use of botulinum neurotoxin (BoNT, Botox) produced by *Clostridium botulinum*. It is highly toxic and the causative agent of Botulism, a rare but potentially fatal illness. However, the toxin is frequently used for medical and even cosmetic applications.

Botulinum neurotoxins are a family of Zn^2+^-metalloproteases [76]. The toxins cleave the three SNARE proteins synaptobrevin, syntaxin, and SNAP-25 in mammalian cells. SNARE proteins are required for the fusion of the vesicular membrane with the plasma membrane and therefore for the release of neurotransmitters into the synaptic cleft. Thus, their cleavage leads to neuro-paralysis [77]. Botulinum neurotoxins bind to at least two receptors present in the presynaptic membrane, a polysialoganglioside and a glycosylated synaptotagmin, situated in the luminal membrane of synaptic vesicles, which leads to high cellular selectivity of the neurotoxins [78,79,80]. As a typical AB toxin, the enzyme domain of Botox is released from the acidified endosome into the cytosol, where it cleaves SNARE proteins leading to paralysis of the affected neuron [81,82]. Among the first clinical applications of Botox were spasmodic illnesses like blepharospasm, cervical dystonia, and limb spasticity. The therapeutic use of Botox is now widespread and ranges from tremors and hyperhidrosis to migraine, depression, and chronic pain. However, the main application of the toxin is its use for cosmetic indications [83].

*Bordetella pertussis* adenylate cyclase (CyaA) interacts with the integrin CD11b/CD18 (complement receptor 3) present in cells of the innate immune system, including dendritic cells which are highly sensitive to the toxin. In contrast to other AB toxins, CyaA forms a pore directly in the plasma membrane to translocate the invasive adenylate cyclase [84,85]. Factors for this process seem to be the membrane potential and the activity of calpain, which cleaves off the catalytic part of CyaA [86,87]. The intrinsic immune cell specificity of the pore-forming subunit of CyaA is interesting for the artificial delivery of cancer cell antigens into dendritic cells, ultimately stimulating tumor-specific cytotoxic T-cells [88,89].

## 4. Secreted Injection Systems

To reach their targets within cells, protein toxins need to cross the plasma membrane. In contrast to the mode of action of AB toxins, some bacteria directly inject proteins into mammalian cells with the help of a syringe-like apparatus (Type-III secretion systems). This requires direct contact between the bacterium and the target cell. Bacteria with type-III secretion systems include Salmonella and Yersinia species, which likewise have been used for the injection of heterologous proteins into mammalian cells [90]. A functionally similar injection apparatus is synthesized by the insect pathogen *Photorhabdus luminescens*. A noteworthy difference to the type-III secretion systems is that here, the whole machinery is composed of three isolated proteins (TcA, TcB, and TcC), which are secreted by the bacteria [91,92]. The photorhabdus toxin complex (PTC) assembles autonomously in body fluids or in a cell culture medium (Figure 2). This attribute allowed recombinant expression in non-pathogenic bacteria in sufficient amounts for purification [93,94]. Structural analysis of the large (1.4 MDa) toxin complex (TC) by cryo-electron microscopy showed that the A component forms a pentamer with a needle-like structure surrounded by a shell building up the injection apparatus. TcB and TcC monomers assemble into a cocoon-like structure enclosing the protein to be injected (BC). The enzymatically active part of PTC is located within the C-terminus of TcC. It is proteolytically cleaved inside the cocoon by an intrinsic protease and is loaded into the syringe after BC binds to the A-pentamer [95,96]. Similar to *Bacillus anthracis*, *Photorhabdus luminescens* expresses several C components with diverse enzymatic activity. It was possible to exchange this part for diverse cargo proteins that are similar in size and charge. Using such engineered PTCs as transport units for the delivery of heterologous proteins into cells allowed the loading of more than 10^5^ cargo proteins into each cell with a coverage close to 100% of injected cells [94,97].

Recently, a second extracellular injection system with a phage-tail like architecture has been isolated from *Photorhabdus asymbiotica* [98]. Photorhabdus virulence cassette (PVC) is the gene cluster encoding more than 20 different proteins. These assemble into a contractile injection system composed of a base plate complex, a phage tail-like structure, and a cap. The recombinant nanosyringe could also be manipulated for the injection of several cargo proteins [99]. Compared to PTC, the charge and size of the cargo is less restricted. Proteins up to 130 kDa have been successfully injected. However, the much larger size of the injection system may be a disadvantage when considering tissue penetration.

## 5. Conclusions

The broad knowledge about bacterial protein toxins, their mode of action, cellular receptors, membrane insertion motifs, pathways into the cytosol, substrate specificity, and selectivity together with structural information has allowed their application as cell-biological and even pharmacological tools. Because of their ability to enter mammalian cells, bacterial toxins have been engineered to transport heterologous cargo into cells. Moreover, some of them have been re-targeted toward specific cells and tissues.

Without a doubt, the tight substrate specificity of the bacterial toxins with enzymatic activity is extremely useful for the analysis of target protein-dependent signaling pathways and functions. One advantage of toxin utilization is that there is no need for over-expressing dominant negative or constitutively activated mutant signaling proteins. This initially preserves their physiological pool without the need for overexpression. However, degradation and re-expression of the toxin-modified proteins have been described [100,101,102]. A further benefit of using toxins is that, depending on the toxin concentration, up to 100% of cells can be targeted. This improves experimental methodologies. For clinical applications, the formation of antibodies against foreign proteins must be expected and may limit the benefit following repeated toxin exposure. Remarkably, the frequently used Botulinum Neurotoxin BoNT/A1 appears to be a poor antigen. Very few patients develop neutralizing antibodies, even when the treatment is continued for years [103,104]. Such promising examples of bacterial toxins successfully used as drugs enhance the probability of developing further toxin-based reagents for clinical applications.

## Figures and Tables

**Figure 1 toxins-16-00202-f001:**
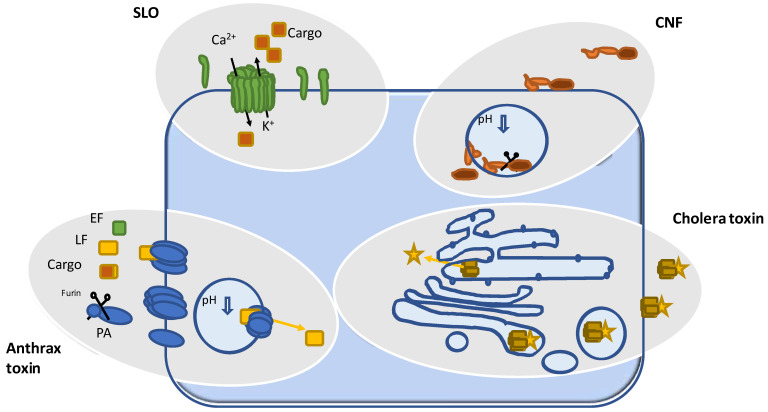
Toxins used as tools. SLO: The pore-forming toxin Streptolysin O binds to cholesterol-rich domains and inserts into the plasma membrane of mammalian cells. It oligomerizes to form water-filled pores through which ions and small molecules can enter cells by diffusion. CNF: Cytotoxic necrotizing factors are taken up by receptor-mediated endocytosis. Following proteolytic cleavage, the catalytic domain is released into the cytosol. This process requires acidification of the endosomes. In the cytosol, CNFs activate Rho proteins by deamidation. Cholera toxin: The toxin is composed of a cell-binding pentamer and an associated ADP-ribosyl-transferase. Toxin-containing endosomes are transported to the ER via the Golgi apparatus. From here, the transferase is released into the cytosol by the outwards transporter of misfolded proteins (endoplasmic-reticulum-associated protein degradation, ERAD). Cholera toxin was mainly used to study this transport route. Anthrax toxin: Anthrax protective antigen (PA) binds to the surface of mammalian cells. Cleavage by furin is required for oligomerization to a heptameric prepore. This allows the binding of lethal factor (LF), edema factor (EF), or an engineered cargo. The complex is then taken up by receptor-mediated endocytosis and released from the acidified endosome into the cytosol.

**Figure 2 toxins-16-00202-f002:**
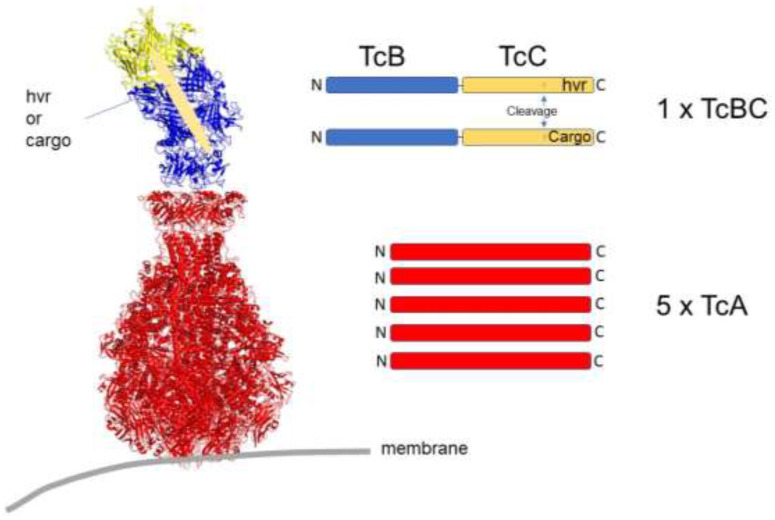
Photorhabdus luminescens toxin complex (PTC) is composed of three isolated proteins (TcA, TcB, and TcC), which are secreted by the bacteria. The A component (red) forms a pentamer with a needle-like structure surrounded by a shell (Adapted from [92]). TcB (blue) and TcC (yellow) monomers assemble into a cocoon-like structure enclosing the protein to be injected. The enzymatically active part of PTC (hypervariable region, hvr) is located within the C-terminus of TcC. It is proteolytically cleaved inside the cocoon by an intrinsic protease, as indicated. The engineered protein TcBC is a fusion of TcB and TcC. The catalytic hvr can be exchanged by diverse cargo proteins. x = times.

**Table 1 toxins-16-00202-t001:** Selection of secreted bacterial toxins that target mammalian cells: Pore-forming toxins (top), toxins with enzymatic activity (bottom).

Pore-forming toxins (selection)				
**Toxin**	**Bacterium**	**Class**	**Receptor**	
Aerolysin	*Aeromonas* spp.	β	CD52	
Cytolysin A	*Escherichia coli*, *Salmonella enterica*	α	Cholesterol	
ε-Toxin	*Clostridium perfringens*	β	HAVCR1	
Hemolysin α/γ	*Staphylococcus aureus*	β	Phosphatidylcholin/ADAM10	
Hemolysin BL	*Bacillus cereus*	α	Cholesterol	
Listeriolysin	*Listeria monocytogenes*	β	Cholesterol	
Perfringolysin O	*Clostridium perfringens*	β	Cholesterol	
Streptolysin O	*Streptococcus pyogenes*	β	Cholesterol	
Toxins with enzymatic activity (selection)				
**Toxin**	**Bacterium**	**Enzymatic Activity**	**Substrate**	**Protein Receptor**
Anthrax toxin (EF)	*Bacillus anthracis*	Adenylylcyclase	ATP	TM8, CMG2
Anthrax toxin (LF)	*Bacillus anthracis*	Protease	MAPKK, Nlrp1	TM8, CMG2
Botulinum neurotoxin	*Clostridium botulinum*	Protease	SNARE proteins	Synaptotagmin
C2 toxin	*Clostridium botulinum*	ADP-Ribosyltransferase	Actin	
C3 toxin	*Clostridium botulinum*	ADP-Ribosyltransferase	RhoA	Vimentin
Cholera toxin	*Vibrio cholerae*	ADP-Ribosyltransferase	Gα s	
Cytotoxic necrotizing factors 1,2,3	*Escherichia coli*	Deamidase	Rho GTPases	
Cytotoxic necrotizing factor Y	*Yersinia pseudotuberculosis*	Deamidase	Rho GTPases	
CyaA	*Bordetella pertussis*	Adenylylcyclase	ATP	CD11b/CD18
Dermonecrotic toxin	*Bordetella bronchiseptica*	Transglutaminase	Rho GTPases	Cav3.1
Diphtheria toxin	*Corynebacterium diphtheriae*	ADP-Ribosyltransferase	Elongation-factor	
Pertussis toxin	*Bordetella pertussis*	ADP-Ribosyltransferase	Gα i	
Pasteurella multocida toxin	*Pasteurella multocida*	Deamidase	Gα subunits	LRP1
Shiga toxin	*Shigella dysenteriae* *Escherichia coli*	Depurinase	RNA	
Toxin A	*Clostridoides difficile*	Glucosyltransferase	Rho GTPases	LRP1
Toxin B	*Clostridoides difficile*	Glucosyltransferase	Rho GTPases	LRP1

Abbreviations: ADAM10: disintegrin and metalloprotease domain-containing protein 10, Cav3.1: T-type calcium channel, CD: cluster of differentiation, CMG2: capillary morphogenesis gene 2, HAVCR1: hepatitis A virus cellular receptor, LRP1: LDL-receptor related protein 1, MAPKK: mitogen-activated protein kinase kinase, Nlrp1: NACHT, LRR, FIIND, CARD domain, and PYD domains-containing protein 1, TM8: tumor endothelial marker 8.

## Data Availability

Not applicable.

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
