# Peer review of "Some Examples of Bacterial Toxins as Tools"

_toxins, 2024, doi:10.3390/toxins16050202_

Round 1
Reviewer 1 Report
Comments and Suggestions for Authors
See attached document.

There are numerous small langage errors that should be addressed. The naming of some of the toxins is also incorrect, but this is addressed in my review.
Author Response
We thank the reviewer for his/her effort and the suggestions made. We changed the manuscript as indicated by red color in the new version. We think that the manuscript improved by the changes made.
Line 4 – host’s defenses would be a better description of how several of these toxins work rather than
“host’s immune system”.
We changed the wording accordingly.
Lines 7-8 – change to “activities are diverse and range from”
We changed “reached” to “range”
Section 1:
Line 32 – “and also” is redundant. Delete also. Delete ‘also’ on line 47 and line 168, as well.
We deleted “also” three times.
Throughout the entire manuscript, toxins are referred to incorrectly. Choleratoxin should be cholera
toxin (and cholera does not need to be capitalized). Shigatoxin should be Shiga toxin (and Shiga is the
name of a discoverer, so it should be capitalized).
Anthrax is a disease, not a toxin. When referring to the toxin, the authors need to say “anthrax toxin”,
not anthrax.
In Table 1, the Corynebacterium diphtheriae is spelled incorrectly.
We corrected the spelling of the toxins.
Line 66 – Cholesterol does not need to be capitalized when not at the start of a sentence.
We corrected the spelling.
Line 90 to 97 – There is a section starting with “The alteration of the cleavage …” and ending with “…the
factor of 100.” is out of place. The overall sentence seems to be describing the anthrax toxin
mechanisms, but this short section refers to potential uses of the toxin. The use of the toxin should
perhaps be a separate paragraph that follows description of the mechanism.
To make this section more clear and separate it from the pure toxin description, we added:
“This property has been utilized for toxin engineering:”
Line 92 – “This allowed to turn the toxin’s” should be “This allowed turning the toxin’s”.
The wording was changed as suggested.
The paragraph beginning on line 112 is about Clostridium botulinum C2 toxin. Because of this, the
authors should check to see if the term CT-1 on line 120 is correct. They may mean C2-I since CT usually
refers to cholera toxin in the literature and for cholera toxin, it is the CT-A1 subunit that has ADP
ribosyltransferase activity.
This was a typing error. We mean C2-I, not CT-1. This is corrected in the new version of the manuscript.
Line 139 – “facilitated” might read better if changed to “facilitates”.
The wording was changed as suggested.
Lines 142 and 145 – The authors should use “cell biology” rather than “cell biological”.
The wording was changed.
Lines 148 to 152 – It is not clear what message the authors are trying to convey here. As written, this
short section does not make sense for this reviewer.
The section was removed.
Line 156 – Golgi should be capitalized.
This was corrected.
Line 155 and elsewhere – Pertussis toxin, not pertussistoxin.
We corrected the spelling of the toxins.
Line 159 – This should read “enzyme is refoleded”.
We added “is”.
Line 169 – “A fact that emerged the idea” should read “A fact from which emerged the idea”.
The wording was changed as suggested.
Line 175 – diphtheriae is not spelled correctly.
This was corrected.
Line 178 – Should read “as a powerful readout…”
We added “a”.
Line 183 – Malignant does not need to be in parentheses.
We removed “malignant”.
Line 205 – As an element, calcium does not need to be capitalized.
This was corrected.
Lines 223-224 – For the statement “This is probably because there is no need for the cell to destroy an
active protein.” have a reference to support it.
This is the personal opinion of the author. We tried to make this clear by the word “probably”.
The paragraph beginning on line 260 can be merged with the one that precedes it.
The paragraphs were merged.
Line 282 and elsewhere – Photorhabdus should probably be italicized.
We think it should not be italicized when it is part of the toxins name.
Line 307 – “information allowed” might read better if it said “information has allowed”.
This was changed as suggested.
Line 312- Should be “target protein” (the space between the words in missing.
We added a space.
Final paragraph (starts on line 323): That paragraph seems to be completely out of place here. If the
authors want to include this toxin, they should dedicate a bit more time to it.
There should be some sort of concluding or summarizing section or paragraph. As written, the
“Outlook” section is inadequate.
We removed the sentence and changed “Outlook” to “Conclusion”.
Reviewer 2 Report
Comments and Suggestions for Authors
I am not sure if Fig. 1 has been removed while the manuscript was being prepared for double-blind review, but there are no figures at all in the document I have been given and the article really needs some figures to illustrate key concepts throughout. There are also some structural issues with the way the work is presented. For example, the abstract seems to be doubling up as an introduction; section 1 and 2 are attempting to introduce classes of toxin, but they don’t introduce all of the toxin classes that are discussed. Section 3 is out of line with the previous two sections as it now starts to classify the toxins by their enzyme activity. The title of this section does not really fit with the content: the title implies the section will be about understanding the enzyme activity, but this is really the most important part of the review which is describing the applications of the toxins – i.e “toxins as tools” which is the title of the manuscript. Some things appear to be missing – for example cholera toxin B-subunit is used in the clinically available Dukoral vaccine. There have also been several very interesting re-engineered BoNTs – for example from Bazbek Davletov’s group.
Line 8 – ribosyl or glycosyl-transferase activity – should this be glycosidase activity?
Line 9 (Tab.1) – is there a table? It should not be referred to in the abstract
Line 45 – where is Fig.1? There are no figures in the manuscript I have been sent to review
Lines 51-53 – cholera toxin and shiga toxin are not pore-forming toxins – they mediate endocytosis of their toxic enzyme and its trafficking to the ER where they cross into the cytosol through an endogenous pore – it is not part of the toxin. They should not appear in section 1. there doesn’t appear to be any reference given for shiga toxin.
lines 89-90 – is the pore pentameric or heptameric?
lines 87-93 – it would be good to explain what the enzyme activity of botulinum toxin is
Line 94 – What is CT-1? – it has not been defined
Line 105 – section 3 switches to understanding the enzymatic activity of bacterial toxins, but at this stage the different types of toxin have not all been introduced. For example the cholera toxin family needs to be described as a separate section for non-pore-forming toxins.
line 128-129 and throughout the manuscript. Cholera toxin and pertussis toxin each have two words – i.e. not choleratoxin
Line 139 – C. botulinum C3 toxin is now described, but it was not mentioned in the previous section that it exists – only C2 toxin was described. This makes it very difficult for the reader to get an overview of all of the toxin families.
line 144 – If C3 lacks a binding or translocation domain, then how does it act as a toxin? Are the C2 and C3 abbreviations domains or toxins here?
Line 147 – what is vimentin? please explain.
line 254-263 and throughout the manuscript – it would be really helpful to have some figures – for example in line 254 there are three proteins listed (TcA, TcB and TcC), but it is difficult for the reader to picture how these three proteins come together to make the type III secretion system. I would really like to see figures throughout the manuscript – especially where there is structural information for the toxins.
270-277 – same comment as above – figure, please.
lines 287-288 - “This initially preserves their physiological pool.” I am a bit lost here – please can you expand the sentence to explain what you mean?
Comments on the Quality of English LanguageEnglish is good - my only comment was: line 223 – “respectively” is not needed here – if you were describing a list of three specific BoNTs and their targets in the same order it would be appropriate.
Author Response
We thank the reviewer for his/her effort and the suggestions made. We changed the manuscript as indicated by red color in the new version. Most criticism of this reviewer is due to the incorrectly transmitted manuscript (Of course the manuscript is difficult to understand if explaining figure and table are missing). We apologize for the inconvenience.
The intention of this review is to show the potential of bacterial toxins to be used for cell biology. We only want to present examples here and not to cover all toxins known. To address this more directly, we changed the title to: “Bacterial Toxins as Tools – Some examples”
Line 8 – ribosyl or glycosyl-transferase activity – should this be glycosidase activity?
This is indeed a transferase activity.
Line 9 (Tab.1) – is there a table? It should not be referred to in the abstract
Line 45 – where is Fig.1? There are no figures in the manuscript I have been sent to review
Figure and table are given.
Lines 51-53 – cholera toxin and shiga toxin are not pore-forming toxins – they mediate endocytosis of their toxic enzyme and its trafficking to the ER where they cross into the cytosol through an endogenous pore – it is not part of the toxin. They should not appear in section 1. there doesn’t appear to be any reference given for shiga toxin.
A reference for Shiga toxin was added.
lines 89-90 – is the pore pentameric or heptameric?
The pore is heptameric as written. A reference was added.
lines 87-93 – it would be good to explain what the enzyme activity of botulinum toxin is
This should be clear now, given in the table.
Line 94 – What is CT-1? – it has not been defined
This was miss-spelled and is now corrected.
Line 105 – section 3 switches to understanding the enzymatic activity of bacterial toxins, but at this stage the different types of toxin have not all been introduced. For example the cholera toxin family needs to be described as a separate section for non-pore-forming toxins.
We think that this issue is due to missing of the table in the first, incompletely transmitted manuscript. However, we added: “Similarly, non-pore forming toxins have been used for specific labeling of cellular structures. “ to make this more clear.
line 128-129 and throughout the manuscript. Cholera toxin and pertussis toxin each have two words – i.e. not choleratoxin
We corrected the spelling of the toxins.
Line 139 – C. botulinum C3 toxin is now described, but it was not mentioned in the previous section that it exists – only C2 toxin was described. This makes it very difficult for the reader to get an overview of all of the toxin families. line 144 – If C3 lacks a binding or translocation domain, then how does it act as a toxin? Are the C2 and C3 abbreviations domains or toxins here?
We think that this issue is due to missing of the table in the first, incompletely transmitted manuscript.
Line 147 – what is vimentin? please explain.
Vimentin is a the type III intermediate filament protein expressed in all mammalian cells. The protein was also found at the cell surface. An explanation about vimentin is given in the new manuscript.
line 254-263 and throughout the manuscript – it would be really helpful to have some figures – for example in line 254 there are three proteins listed (TcA, TcB and TcC), but it is difficult for the reader to picture how these three proteins come together to make the type III secretion system. I would really like to see figures throughout the manuscript – especially where there is structural information for the toxins.
270-277 – same comment as above – figure, please.
lines 287-288 - “This initially preserves their physiological pool.” I am a bit lost here – please can you expand the sentence to explain what you mean?
We intended to say, that the direct activation of a protein by a bacterial toxin leads to changes in cell signalling without the need of overexpression of the protein. The amount of proteins (the physiological pool) stays constant. We extended the sentence accordingly.
Round 2
Reviewer 1 Report
Comments and Suggestions for Authors
This manuscript has been revised adequately.
Author Response
Thank you
Reviewer 2 Report
Comments and Suggestions for Authors
The authors have made improvements throughout the manuscript. It is helpful to now see Table 1 and Figure 1 which were not included previously.
Table 1 would benefit from having more information on receptors - the authors have focussed exclusively on proteins that act as receptors, but many of these toxins have glycolipid receptors instead of, or in addition to, the protein receptors. Please add them to the table.
In the part of Table 1 describing Toxins with enzymatic activity, some of the toxins are named in full and others are only given abbreviations - e.g. cholera toxin vs Stx (Shiga toxin). Please be consistent in how the information is presented.
Figure 1 is helpful for the toxin that are included in it, but the following two comments from my previous review regarding PTC and PVC have not been addressed by the authors in their response. Please add another figure to address my previous comments:
line 254-263 [now 277-286 in revised manuscript] and throughout the manuscript – it would be really helpful to have some figures – for example in line 254 there are three proteins listed (TcA, TcB and TcC), but it is difficult for the reader to picture how these three proteins come together to make the type III secretion system. I would really like to see figures throughout the manuscript – especially where there is structural information for the toxins.
270-277 [now 293-300 in revised manuscript] – same comment as above – figure, please.
Author Response
Table 1 would benefit from having more information on receptors - the authors have focussed exclusively on proteins that act as receptors, but many of these toxins have glycolipid receptors instead of, or in addition to, the protein receptors. Please add them to the table. In the part of Table 1 describing Toxins with enzymatic activity, some of the toxins are named in full and others are only given abbreviations - e.g. cholera toxin vs Stx (Shiga toxin). Please be consistent in how the information is presented.
We changed the toxins names to their full names in the table. However, we think that the table would be too unclear when adding further toxin receptors and therefore focused on protein receptors as indicated in the table.
Figure 1 is helpful for the toxin that are included in it, but the following two comments from my previous review regarding PTC and PVC have not been addressed by the authors in their response. Please add another figure to address my previous comments:
line 254-263 [now 277-286 in revised manuscript] and throughout the manuscript – it would be really helpful to have some figures – for example in line 254 there are three proteins listed (TcA, TcB and TcC), but it is difficult for the reader to picture how these three proteins come together to make the type III secretion system. I would really like to see figures throughout the manuscript – especially where there is structural information for the toxins.
270-277 [now 293-300 in revised manuscript] – same comment as above – figure, please.
We added a new figure 2 to explain the assembly of the Photorabdus luminescens toxin complex. And the engineered injection apparatus.